# Early Support for People Who Hear Voices: Exploratory Research on Family Medicine Physicians’ Clinical Practice and Beliefs

**DOI:** 10.3390/bs14050357

**Published:** 2024-04-25

**Authors:** Antonio Iudici, Giulia Alecu, Maria Quarato, Jessica Neri

**Affiliations:** 1Department of Philosophy, Education, Sociology and Applied Psychology of Padua (FISPPA), University of Padua, 35139 Padua, Italy; antonio.iudici@unipd.it; 2Institute of Psychology and Psychotherapy (Scuola Interazionista), 35100 Padua, Italy; giulia.alecu@gmail.com (G.A.); quarato.maria@gmail.com (M.Q.); 3Ediveria Center, Zentagasse 18 top ¾, 1050 Vienna, Austria

**Keywords:** family medicine physician, hearing voices, qualitative research, health promotion

## Abstract

Nowadays the phenomenon of hearing voices represents a very fertile and discussed field of research. In psychological and psychiatric fields, the phenomenon has been described as a normal phenomenon, but also as a prodromal stage and as a symptom of psychosis. Through a qualitative research methodology, the aim was to explore how family medicine physicians configure the phenomenon and its clinical and interactive implications. The present research involved 35 family medicine physicians as figures of primary importance in the approach toward people who start to hear voices. Semi-structured interviews have been used and they have been analyzed by the method of discourse analysis. The results show a remarkable difficulty in understanding the phenomenon in all its complexity and the tendency to consider it a symptom or a prodromal stage of psychopathology. Increasing the knowledge of doctors on the subject is necessary so that their evaluation and choice of intervention match the needs of each patient. We also discuss the importance of promoting the knowledge of the potential meanings taken on by the voices in the context of the personal and family background of the individual hearer, and of collaboration with other relevant professionals and services.

## 1. Introduction

In psychological and psychiatric disciplines, hearing voices and related phenomena have been described for a long time mostly as a prodromal stage and as a symptom of psychosis. During the last few decades, the scientific community has been particularly interested in the phenomenon of hearing voices, considering it a normal phenomenon [1,2,3,4,5]; numerous studies have shown that hearing voices is not necessarily something pathologic but that it can actually be functional for the person who experiences it [6].

This new awareness has led also to the editing of the DSM’s section about the criteria for the diagnosis of schizophrenic spectrum disorders; in its recent version, hearing voices no longer appears as an elective symptom for such a diagnosis. This is due to a change that affects not only the reading of this phenomenon but also the related diagnostic category in the manual and the approach developed with its latest version, one that transforms the previous diagnostic category of schizophrenia and any other type of psychotic disorder into schizophrenia spectrum disorder (or any other type of psychotic disorder) according to a dimensional approach. In this sense, this is related not only to the possibility that the so-called symptom of hearing voices may belong to the realm of normality, but also to the management of the relevant heterogeneity and labile boundaries of the previous diagnostic categories in the manual [7].

The development of diagnostic categories has seen numerous changes over time, such as those highlighted above. In addition, it is specified in the manual how issues related to culture and socioeconomic and cultural factors are considered in the diagnostic process. This highlights how cultural differences, for example, can be considered with respect to what may appear to the clinician as delirium [7].

Although this is an aspect that should not be underestimated, what can be highlighted through other studies and research is how hearing voices, in particular, is not an objective phenomenon that possesses cultural characteristics that can be evaluated sharply, as the characteristics of its diagnosis would require, but a phenomenon that is generated through language and culture [8,9,10]. So, this kind of research allows the phenomenon to be questioned and considered as not exclusively associated with symptomatic or prodromal conditions, but with normal and not necessarily clinical experiences that can be more or less competently managed and evaluated according to each person’s specific needs [4,11,12,13].

Starting from this premise, scientific studies began to be interested in those who experience this phenomenon in positive ways, exploring the benefits that can be derived from this condition, which can be defined as a specific cognitive function [10,14,15] or rather called a “capacity” [16]. Among the positive aspects identified, we can mention the function of the voices as a source of encouragement and a way to avert feelings of loneliness [1], encourage personal growth [17], and provide support in critical moments [5]. The role of factors that contribute to making the phenomenon of hearing voices a positive experience for the person has been explored; for example, the frequency of the voices, their content, and coping strategies are important [18], but of particular importance is the quality of the relation between the person and their voices, in addition to their beliefs about it [19]. In this regard, the perception of being able to control the voices seems to favor a positive experience about it [20,21,22].

Although the scientific opinion about this phenomenon has been changing over time, in popular thinking and often even among health professionals, this change seems as if it has entirely not happened [23,24]; one who hears voices often is still reputed to be a victim of deep illness, labeled as problematic and considered socially dangerous [25,26] according to the traditional medical and psychiatric conceptions about hearing voices specified above.

The belief and the reductionist conception that links hearing voices to a prodromal stage or a symptom of psychosis risks developing stigmatizing and discriminatory approaches towards the people who hear voices, which also undermines their serenity and ability to share their experiences with health professionals [27,28,29,30,31].

Indeed, many people who hear voices tend to isolate themselves in fear of other’s people judgment, minimizing the possibility of receiving support for their condition by meeting sources of normalization [17].

Nevertheless, thanks to the trusting relationship between doctor and patient, the family medicine physician, also known as the general practitioner, seems to be the person who can still intervene in this type of phenomenon when it starts to appear. This type of professional represents the first healthcare practitioner for the one who has an unexplained psychological experience [32].

For this reason, the scientific community became interested in the quality of treatment offered by family doctors to patients who turn to them for psychiatric problems. However, in the studies conducted, doctors declare themselves ill prepared for facing this type of request, admitting their difficulty in enhancing their service [33].

In a recent study, doctors have confirmed this problem by identifying the possible causes of this as the lack of time and resources, low trust in their ability to handle complex mental illness, and the difficulty in referring the person to another specialist [34].

The criticality revealed by these studies reports a risk that it is better to not underestimate: the doctor may choose to manage the patient independently by administering psychiatric drugs without consulting a specialist. Different studies in the scientific literature show the inadequacy of the drugs prescribed by family doctors who do not have sufficient training, as they fail to produce the desired effects [35,36].

Furthermore, the employment of some psychiatric medication implies in itself a few collateral effects that might be mistaken for an aggravation of the problem [37]. Some medicines do not have a good response rate, not even when the treatment is prescribed by a psychiatrist; for example, antipsychotics do not elicit a satisfying drug response in schizophrenic patients and cause at the same time disabling collateral effects, which in the long term seems to compromise irreversibly the social functioning in the person [38,39].

What can be pointed out from the available literature is how there are no definitive data concerning the efficacy of drugs, such as different generation antipsychotic drugs, in relation to the diagnosis of schizophrenia spectrum disorder. There are numerous data regarding poor outcomes and re-hospitalization [40], a long series of side effects that is also related to low adherence to drug therapy, stigma, and/or iatrogenic effect [41,42,43,44].

These critical aspects related to the medical intervention and efforts by the physicians highlight the presence and impact of the traditional medical and psychiatric conceptions about the phenomenon of hearing voices. From that emerges the difficulty of managing the complexity of the phenomenon in terms of observation, evaluation, and intervention for what the person’s needs may be in the face of the peculiar individual experience.

Recent research, however, has demonstrated the efficacy of psychological treatment, even for patients reputed to be particularly compromised. In the case of people who hear voices, who often are considered schizophrenic, several treatments have proven effective, the main among these being the compassion-focused approach, acceptance and commitment therapy, and group psychotherapy [3,23,45,46,47,48,49,50,51,52,53,54,55,56]. With psychotherapy, those who hear voices can learn strategies for the management of voices and have a positive experience of the phenomenon, without suffering the collateral effects of medicines [44,57,58,59,60,61,62].

Despite this, studies recently conducted in Italy [63,64] have proven how only 13% of healthcare professionals involved (N = 264) consider psychotherapy for the treatment of patients deemed schizophrenic, still considering psychiatric drugs as an elective and essential treatment.

With regard to people who hear voices, no research has yet explored the ways in which the family medicine physician, as a crucial figure in primary care, responds to the needs of the person. We asked ourselves the following questions: How do they conceive of the phenomenon of hearing voices? How do family doctors take charge of the voice-hearing patient? What type of health interventions do they know, and how do they evaluate the most appropriate one?

The aim has been to discern how primary care doctors configure the phenomenon and the voice hearer, and how they manage the intervention to promote health. Indeed, we are interested not only in the choices for handling these patients, but also in the criteria that establish these choices and the reflections that generate them.

This research makes it possible to shed light on the conceptions and beliefs used by physicians, as well as the criteria and procedures of intervention and their clinical implications. This can be useful to offer additional elements and tools to make reading and intervention more effective in this field and in relation to the needs of the people involved.

### Aims

The research questions turned into the following objectives:(1)To investigate how the phenomenon of hearing voices is configured by the physicians, specifically, what conceptions and beliefs are held about the hearers and their family members;(2)To investigate how the management of a voice hearer occurs, focusing on intervention and referral to specialists.

## 2. Methods

### 2.1. Research Team and Reflexivity

The research team consists of a professor and researcher from the University of Padua (man), two collaborators and lecturers from the School of Interactionist Psychotherapy, and a psychologist-psychotherapist (woman). One of these collaborators manages a center dedicated to psychological counseling for hearing voices. All of the people involved are also psychologist-psychotherapists. The whole research group is united by participation in research and clinical activities with respect to the topic of hearing voices according to the interactionist perspective.

The research design was constructed through a collaborative practice by the research group; in particular, A.I. was involved in conceptualizing and supervising the entire project; G.A. mapped potential participants, managed the interviews and the data analysis, and produced the initial results; M.Q. participated in conceptualizing the project and supervising the data analysis; and J.N. collaborated and supervised the research methodology and its use.

In all moments involving interaction with the participants, contacted for the first time through the research, anticipations were constructed with respect to possible critical issues and resources emerging from the interaction (thus involving the interviewer, but also other group members and their roles) and potential ways of management developed. For the management of the entire research design, as well as that of all interactive elements anticipated or assessed as relevant (e.g., gender, age, theoretical and disciplinary background of the interlocutors, and the peculiar theme of the research), the research objective, the possibility of fostering the exchange and emergence of elements relevant to the participants, as well as the constant shared analysis among the group members, was kept as a reference.

### 2.2. Methodological Orientation

This study bases its assumptions on interactionism [65,66,67]. This theory assumes that humans are active subjects, whose actions are intentional and meaning-oriented; the meanings are constructed by interactive processes and modified through an interpretative process by the subject with the environment [68].

Likewise, social roles are delineated through the anticipations and expectations generated in the interactions. Therefore, because humans continuously interact with their surroundings, they are scientifically interested in the design process of themselves and of the world [69,70,71].

This study explores how family doctors configure the phenomenon of hearing voices, by studying the linguistic and discursive processes through which it is built [71]. Language is considered the generative matrix of reality, social roles, and personal identity, socially co-constructed and continuously renegotiated in the interaction with others [72]. This theoretical perspective allows us to consider how the realities under investigation, such as hearing voices in this case, are not definable as given realities, but are generated through the discursive modes that are used to describe them. For this reason, they cannot be considered independently of their socio-cultural reference contexts and social functions. The analysis of the discourses of physicians as figures of primary importance in approaching this type of experience is considered relevant, therefore, in that it represents not only an analysis of opinions or preeminent meanings, but of discursive modes with pragmatic value that guide action and contribute to defining this kind of phenomenon.

Coherently to the theoretical background, a qualitative approach has been considered for understanding the complexity of the experiences provided by the participants, limiting the risk of reducing them to quantitative categories. The research here is understood as a relational, collaborative practice and dialogical process [73,74,75], and the object of analysis is considered co-constructed by the researchers and the interlocutors [72].

The entire research path and the analysis of the qualitative data have been set following the COREQ procedure of control [76].

### 2.3. Eligibility Criteria

To participate, being a family doctor and having previous experience of at least five years in public healthcare services were the necessary inclusion criteria.

### 2.4. Sampling and Sample Size

This study was conducted in Italy, where the family medicine physician is a doctor provided by the national health system (SSN). The family or individual citizen can choose their doctors from those available. The doctor provides first-level assistance in his medical practice, at the home of the patient, and in residential facilities (residential social-health facilities dedicated to elderly people who are not self-sufficient [RSA], Nursing Homes, and Community Hospitals).

The first participants were recruited by phone calls, drawing from the Local Health Unit of Padua’s doctor list, continuing with the “snowballing” technique. In particular, each participant involved was asked for knowledge of potential interested colleagues and willingness to share the research project and the researchers’ contacts with them. In this way, researchers were also contacted by other professionals, who proceeded in the same way.

The research involved 35 family medicine physicians working in Italy, 18 men and 17 women, with an average age of 53 years and an average professional experience of 21.5 years. Demographic data are presented in Table 1.

### 2.5. Setting

The research purpose was briefly explained to each participant, and any doubt about the research methodology was clarified. A module was also offered to each participant so they could provide their informed consent. Our study was approved by the University of Padua’s ethics committee for research. The interviews lasted on average 45 min and were conducted remotely in electronic form, recorded, and integrally transcribed. The researchers have emphasized the need to use an audio-recording tool to produce as faithful a transcription of the interview as possible.

### 2.6. Data Collection

The semi-structured interview has been used as a research instrument and method for data collection. This type of interview has been chosen to gain access to the symbolic and experiential universe of the family physicians interviewed and to grasp those meanings regarding the specificity of the context in which they are placed [77].

This type of interview involves the use of a track (Table 2), used by the interviewer to guide the conversation towards the research objectives. However, it preserves a good degree of flexibility, allowing for supplementary questions to be asked to deepen interesting or unclear conceptions or beliefs, and the interviewees are completely free to express themselves, without limits of time or content [78,79].

With this kind of interview, the two objectives in Table 2 were pursued. On the one hand, the text was collected to delve into the configuration of the phenomenon, i.e., conceptions and beliefs about people who hear voices and family members and their implications. On the other hand, the text was collected to delve into how these situations are taken care of by physicians, i.e., how intervention and referral to other professionals are handled. The questions formulated served as a guide, allowing the adherence to the objectives outlined, but at the same time, leaving room for other insights, clarifications, and specifications. For these reasons, this interview is here understood as a method and not just as a tool since it is guided by the researcher’s evaluations and choices, that is, the research design and theoretical-methodological references.

### 2.7. Data Analysis

The textual corpus was analyzed by discourse analysis, which is a set of qualitative research methods aimed at identifying how a lived experience is discursively built [80] and from which the possibilities of action can be grasped [81]. The analysis of discourse allows us to understand how people orient themselves concerning a particular issue or theme. By regarding the discourse as practice-oriented to the situation [82], we have chosen the conversational positioning of Harré et al. [83] as a specific method of analysis. This process of analysis is not individual but is affected by speeches and narratives present in the cultural matrix of reference.

The researchers studied the text, starting from the transcripts and trying to identify the main ways in which the doctors interviewed positioned themselves relative to the areas of investigation.

The first analyses were carried out individually by the authors of the research, and then these were cross-compared, highlighting possible doubts in the reading of the processes detected.

Once we found the fundamental discursive registers concerning positioning, we systematized the work in processual categories to share the meaning and facilitate the reading of the results. Both positionings and processual categories, in accordance with the methodology of analysis, are understood as discursive modes that have valence in contributing to the generation of the phenomenon under investigation.

Considering the specific objectives of the research and what was analyzed from the texts of the professionals interviewed, we therefore divided the results into three processual categories: (1) how hearing voices and the voice-hearers are configured; (2) how the case intake and the referral to another specialist is managed; and (3) how referral criteria are used. For each of these categories, the main positionings were identified through which participants configure the phenomenon and the voice hearer, and how they manage the intervention to promote health.

### 2.8. Quality

Regarding the evaluation and management of the quality of the research, the epistemological, methodological, and theoretical references that defined the research design were taken into account. Thus, efforts were made to ensure consistency between these, and the choices and the methodological steps made by the researchers, making what was collected and the constructed dialogue consistent with the analysis of the relevant literature. At the same time, data were collected and analyzed with constant reference to their context within a situated and processual dimension of reality construction [76]. This was consistent with the conceptual background used.

## 3. Results

### 3.1. How Hearing Voices and the Voice-Hearers Are Configured

This section presents the main positionings related to the first processual category identified through the text analysis. The results are summarized in Table 3.

#### 3.1.1. Absolutizing the Hearing of Voices as a Psychopathological Symptom

Among the interviewed doctors, the belief is strongly present that hearing voices is a psychopathological symptom. Often this idea is expressed without leaving any space for doubts or alternative hypotheses.

The phenomenon is tendentially associated with schizophrenia, psychosis, delirium, or dissociation disorders, always giving it a negative and invalidating character, following a generalization process: *“Eh, beh, a schizophrenia, eh, it is a greater psychosis to better define it (…) and it is very devastating for the patient because it influences his whole life(…) they have not a life anymore”; “ It is a psychosis problem, It is a delusional problem”; “Clearly, a person who hears voices is blocked, I mean it is a damage for his/her life”.*

Additionally, in some cases, the voice-hearers are deemed socially dangerous; in these prospects of the phenomenon, the voice is considered able to control the person, who is dictated by it and loses the capacity to renounce sometimes violent behaviors: *“(The voice listeners acting) depends on what the voice says to them; they don’t own a life anymore, but they live according to what is told to them, and it is dangerous!”; “ The (voice listener) act not apparently- according to his/her desire but because he/she perform commands that are always negative, violent, that’s it”; “ It is a disorder (…) that in some cases could also lead to doing gestures that conversely would not be accomplished!”*

To a lesser degree, some answers are defined through a possibilistic modality referring to hearing voices as an experience not stackable to psychopathology; some speculate that the experience of hearing voices is not necessarily a negative one, while others interpret this as a possible strategy put in place to face critics or stressful moments.

What is possible to highlight from this kind of positioning is thus the sanctioning and affirmative attribution of the experience of hearing voices to some form of psychopathology. There is present, although to a lesser extent, the contemplation of the possibility that this experience may also be a resource as well as a prodromal stage or as a symptom of psychopathology.

#### 3.1.2. Diagnosing the Patient Based on a Hypothesis Derived from Personal Experience

A fundamental aspect that connects lots of discourses is to delineate the phenomenon and the experiences accompanying it, based on the professional experience gained. Often doctors use hypothetical and experiential ideas to define reality. They state that the answers provided are based on one or more episodes, in which they provided initial assistance to people who hear voices: *“Probably, it is a symptom, we can say, referred by some of my patients, maybe due to a loss of contact with reality”; “At least, regarding those few experiences that I have had about this particular symptom”; “…Ehm, even here it depends, I have patients (…) whereas in first two patients that I have told you before, eh, hearing-voices is occasional and it doesn’t disturb them too much, in others is more…is more constant, more disturbing. Ok?”*

Instead, in other cases, doctors informed us that they had not gained experience in this field and that they do not know even the “general coordinates”, declaring themselves unprepared and not involved: *“Always in my ignorance about this phenomenon, because that’s it what it is, everyone has to admit their limits”, “It is not my area”, “I don’t have a point of view, I mean I am a doctor”, “The fact that people hear voices inside them? (Long pause) It is something that leaves me a little surprised”.*

What can be highlighted from this type of positioning is the reference to the physician’s personal experience and history to define and understand the phenomenon under discussion, rather than some other element from the medical discipline and/or literature. Such positioning is related to personal beliefs related to unpreparedness, i.e., to not having cognitive elements available to handle and intervene in experiences and situations that refer to hearing voices.

#### 3.1.3. The Voice-Hearers as a Cause of Suffering for Their Family

In the answers, the voice-hearers are frequently described according to the suffering that they create in their families. The family is often described in a victim role, in which the weight and the practical consequences of the phenomenon, considered almost unmanageable, fall:

*“(The listeners) carry a deep malaise that falls also on who surround them”; “it is as if the family member is suffering a loss, mourning. Truly (…) you lose your mind!”* Indeed, they assume that those who surround the voice-hearers are not able to face the situation, and they can do nothing more than suffer the consequences of that in everyday life: *“He/she is not well received in the family because he/she becomes almost a problem”. “What do I do now? Where do I take him/her? What should I do?” “(…) they are worried and then, living together becomes a tragedy, a complication”.*

This belief is tendentially generated by causal considerations applied to different criteria. First, coherently with what was previously said and assuming it is psychopathology, the reasons are traced according to the idea that hearing voices can generate nothing more than negative experiences, even for those not directly affected: *“No one in the family wants to accept the situation, given that it is about a psychosis”; “Those who surround these persons imagine that this person has issues! And surely, he/she is not perceived as a normal person!”*

Causal reasoning is applied also to criteria associated with the cultural education level of the family, deemed unable to contrast the stigma: *“However with poor education, it gets a little bit harder because, in the end, the person is called crazy. There is more difficulty in accepting, we can say, the pathology”.*

Always in a causative way, a participant supported that the cause of the problem can be the awareness of the problem itself: *“It is concerning for those around him/her because rightly they recognize the anomaly”; “(long pause) Eh, it is a destruction (…) an awful experience and the relatives feel very bad because they notice that something goes wrong”.* The process that generates this causal thinking is thus anticipatory: negative consequences and lack of resources are anticipated, principally generated by personal hypotheses and beliefs.

What can be brought out from this type of positioning is the causal relationship between the experience of hearing voices and subsequent problematic behaviors, as well as various discomforts and social impairments for the person involved and family members. The role of the psychopathology from which the protagonist of the experience is considered to be affected is judged to be a priority, i.e., in most cases, the engine from which the awareness intervention for the individual and family must start.

### 3.2. How the Case Intake and the Referral to Another Specialist Is Managed

This section presents the main positionings related to the first processual category identified through the text analysis. The results are summarized in Table 4.

#### 3.2.1. The Exclusive Referral to the Psychiatrist

The potentially practicable interventions identified by the doctors are different (neurological, psychological, and psychiatric). Nevertheless, the essential use of drugs is often highlighted, coherently with the idea of considering the voices as a psychological sign.

That is reflected in *the assumption* that the intervention of a psychiatrist is strictly necessary, as the only useful professional figure in this case, or the support of other specialists: *“Clearly, I have directed them to a psychiatrist because…I mean however these illnesses have a particular complexity, I can’t treat them by myself as a simple case of anxiety, a simple insomnia”; “you have to send him/her to the psychiatrist, eh! Eh, how do you cure him/her (with a resigned tone)? I mean in psychological terms, you can’t do anything, with counseling you can’t do anything, drugs are needed, eh, absolutely”; “I mean, this is definitely a psychiatric symptom, I don’t know another specialist who can take care of an auditory hallucination”.*

Thus, tendentially, in the first instance, a psychiatrist is consulted; the univocity of referral, which is independent of contextual characteristics, is sometimes justified by the impossibility of doctors thinking about other interventions for people who hear voices: *“But always regarding the DOCTOR, the best specialist to which to send the patient is the psychiatrist, and the psychiatrist will decide, hmm, how the problem has to be managed”.*

Indeed, some doctors report an information gap about other possibilities of intervention and referral: *“I think there may exist (another type of intervention) more relational based or psychotherapeutic, but I have never used them, I don’t even know…”.*

What can be highlighted from this kind of positioning is the almost exclusive reference to the use of the figure of the psychiatrist for support in the management of voice hearers, in accordance with a view of these experiences that attributes them to a prodromal stage or a symptom of a psychopathology. In some cases, the reference is direct and sees no questions, with the belief that the physician and other figures cannot be engaged in this type of situation.

#### 3.2.2. The Autonomous Case Take-over as a Symptom Containment Attempt

Sometimes, it happens that family doctors choose to take care of a patient single-handedly. In these cases, they proceed to prescribe some psychiatric drugs by themselves without consulting a specialist: *“I have decided to proceed before with a try with a pharmaceutical approach and then with a specialist approach, sending the patient to a psychiatric referral”.*

They reported to us that this decision sometimes is made to compensate for the absence of competent services on the territory: *“I can start with minimal doses of neuroleptic because, you know, I have been a doctor for 40 years. I have a superficial experience of psychiatric patients, but I have it. So maybe, (I prescribe) a little bit of haloperidol to slightly contain the situation, also because we don’t have a structure very present on our territory. Here in Abruzzo, there aren’t psychiatric clinics. (…) there is a little gap in psychiatric assistance. (…) It happens that I am forced to prescribe haloperidol as, we can say, as a symptomatic cure, while waiting for a consult with a specialist because it is not so automatic”.*

With competent services, we refer to the public operative unit in which psychologists, psychiatrists, and neuropsychiatrists are employed working in close connection with social services and general practitioners. The service is mainly concerned with diagnosing, caring for, and rehabilitating the patient.

What can be highlighted from this type of positioning is the belief that there are not enough available services on the ground that can be engaged. Therefore, the physician proceeds independently by assessing the situation based on their own experience with the goal of giving immediate support, such as through medication.

### 3.3. How Referral Criteria Are Used

This section presents the main positionings related to the first processual category identified through the text analysis. The results are summarized in Table 5.

#### 3.3.1. Sending to the Specialist According to the Cause’s Origin: Physic or Psychiatric

For selecting the professional figure to whom to send the patient, Family Medicine Physicians typically use the anamnesis collection.

This practice helps them to assume the origin of the phenomenon, physical or psychiatric, and consequently orient the referral. As already said, often the choice falls immediately on the psychiatrist, but sometimes the doctor relies on specialist exams to orient the referral, having more elements available: *“I try to understand if any other type of hallucination is present in association with the auditory hallucination (…) and in general, I try to distinguish if there is a physic or psychiatric base, and so later direct the patient either towards a therapy that resolves the cause, or then on physic base, towards a psychopharmacological therapy that resolves, we can say, the question on a purely psychic base”.*

Based on this, they consider referring the person to a neurologist, if a possible physical cause is found, or to a psychotherapist if they identify a psychic cause. Regarding the referral to a psychologist/psychotherapist, it happens tendentially when the clinical case is overall considered mildly compromised and when there is awareness about the fact that the voices are not real. It is, however, carried out with pharmaceutical support: *“In the situation in which there is a semi-awareness or a total awareness that these voices are not real things but only noises that they hear, then it is possible to intervene, as well with psychotherapy wherever possible or they accept to do it, in association with pharmaceutical therapy”.*

#### 3.3.2. Referral to a Specialist Based on the Idea of the Symptom Severity

As mentioned before, some doctors use the evaluation of symptom severity as the criterion to select the type of intervention to activate. Particularly, we noticed that the referral to a psychotherapist is considered exclusively only if the syndromic frameworks are viewed as slightly serious: *“If there are a collection of symptoms like that, in a so florid presence of non-real content, I believe that is useless to send a patient to a therapist”; “If however in dialogues comes out this,-mmm,-this sufferance and indeed this disease, yes I send him/her to a psychiatrist”.*

The belief underlying these discourses concerns considering the prerogative of a psychotherapist only in the more functional cases, in which the voice comes up occasionally or the person is not particularly disturbed. Often, even in these cases, the involvement of the psychiatrist is favored over intervening simultaneously on the pharmaceutical level: *“Mah, if there weren’t other signs and it is an auditory phenomenon, occasional or not, the right therapy could be the behavioral therapy”; “But I always send and I always use anyway, the consult of the specialist who I want that supervise or manage in general, especially, the pharmaceutical therapy or, if it is necessary, the psychotherapeutic path that the patient might do. I mean, I don’t know, because I have not deepened the topic”.*

#### 3.3.3. Sending the Patient to the Specialist According to Their Age

Another criterion utilized to identify the specialist to whom to send the patient is the age of the subject itself. These data help the doctor exclude some hypotheses regarding the origin of the phenomenon and narrow down the range of possible interventions: *“Of course, if she was a younger patient, the situation would have probably involved more than a neurologist, obviously a psychologist, a psychiatrist clearly (…) In this order, first of all, the psychologist, because if it had been a younger person’s experience, I say, a person less old, obviously a problem of dementia would not have been considered, at least not as the first diagnosis”.*

## 4. Discussion

The attitude of doctors towards voice-hearers is essentially based on macro-generalizations, with a strong negative connotation attributed to the phenomenon and the experience lived by the voice-hearers. The doctors focus their attention on the symptom, using a categorial approach, considering less both the unicity of personal experience and the phenomenological complexity that characterizes it. In this way, the patient’s unique experience may be attributed to a preconceived idea, depriving both the patient and the same doctor of a specific intervention and of the possibility of deepening their overall knowledge of the phenomenon.

The origin of this simplification is indeed reduced (and recognized) by doctors due to the lack of information and education about it. However, we do know how much the role of doctors is relevant in making the first evaluation and in management of the referral coherently with the recognized needs, orienting themselves thanks to their professional competencies.

Moreover, referring to the phenomenon in terms of disease, the message that can be conveyed to the voice-hearers is that the voices are harmful entities that inhabit them and that a cure has to eradicate them; as the studies present in the literature show, this contributes to a negative relationship between persons and their voices, disadvantaging an improvement in their personal experience and the possibility of accessing other cures [19].

Additionally, in cases where doctors infer that the voice-hearers are subservient to the voice, they tend to consider them less and less able to handle the voice. As described in the literature [1,19,20,21], giving power to the voices at the expense of the person implies the idea of a suffered experience, fostering the patient’s feelings of inadequacy, and undermining their sense of self-efficacy. Furthermore, being considered “sick” tends to inhibit the sharing of one’s own experience, encouraging self-isolation and reducing the possibility of engaging with sources of support [17,84].

Considering the results, the diagnostic evaluation strongly depends on the specific experiences lived by the doctors involved, and therefore we have noticed a wide variety of ways to explain hearing voices and, consequently, to face them differently.

Additionally, doctors elaborate on the phenomenon in possibilistic terms, showing awareness in considering their knowledge as limited.

Sometimes, doctors constitute their intervention on their evaluation of the case: they collect the anamnesis and submit the patient to specific questions to obtain a general framework of information that, in the case of hearing voices, often proves ineffective. In some cases, doctors state that they turn to psychiatrists and, in other cases, to psychologists, neurologists, or geriatricians according to different criteria.

In other cases, indeed, given their evaluation, doctors consider the possibility of themselves temporarily managing the client by prescribing a pharmaceutical treatment. These data sustain some studies in the literature [39,42,44] that support the idea that an autonomous administration almost always has negative effects. This research added that this aspect could also compensate for the absence of public services, opening a huge debate on the function of medicine and psychiatry about containment rather than cure. What we particularly want to highlight is the risk of proceeding in an autonomous manner where the personal and experiential conceptions and beliefs of the individual practitioner may be used, rather than shared or sharable practices and procedures to promote the health of the people involved. So, in addition to the risk of using treatment (e.g., pharmacological) in an intervention-oriented way on the individual experience or “symptom” rather than on a more complex assessment of the person’s situation and needs, there is also a risk of fueling the fragmentation of services rather than moving by taking advantage of the expertise of every useful profession.

In this regard, the fact that some doctors manage the referral in a univocal, indiscriminate, and almost “automated” way undermines the operational fluidity of the public health services themselves, fueling a dysfunctional vicious circle. The doctor who orients the referral according to generalizations and not according to the client’s needs increases the probability that the service offered in the first instance will be inadequate. This would imply a significant waste of additional time and energy of public services themselves, thus slowing down the process of health promotion.

Many participants do not know that there are several kinds of health interventions to choose from and often do not consider a possible referral other than a psychiatric one. This confirms that the prejudice that hearing voices is in any case debilitating for the person leads doctors to consider the psychiatric-pharmacological intervention as the only effective one [17,63]. A similar referral can lead to lots of voice-hearers, who do not live an extremely negative experience, consuming drugs unnecessarily, exposing them to a cure with iatrogenic or useless effects [38,39,44,85].

Additionally, studies present in the literature show that a mental illness diagnosis accompanied by the prescription of pharmaceutical treatment can develop a negative experience in the person towards the voice, an auto-stigmatization, a feeling of helplessness regarding the voice, a loss of self-esteem and a tendency to isolation [28], factors that once again not only discourage an improvement in patient conditions, but also contribute to the crystallization of the problem.

Moreover, the belief emerges that the psychotherapist is not the specialist responsible for the treatment of patients with deep psychic suffering, and this belief has been confirmed by several studies conducted over several years [3,46,47,48,49,50,51,52,53,54,55,56].

With regards to the relatives, the interviewee considered those who surround the voice-hearers as another victim of the phenomenon. By assuming that the parents are not able to support the person, the risk is that the doctor would avoid involving them, considering them as “not capable to manage the situation”. Such anticipation may inhibit a process of normalization of the experience in the person’s family environment, and beyond that, entails the lack of activation of a possible resource, since it is not recognized as such.

Respondents often attribute the parents’ difficulties to several factors, for example, the level of education, as if only people of culture could grasp the ensemble of the phenomenon, or they feel that parents lack awareness of the disease and its implications.

Considering the recent literature and the studies produced over several years about the phenomenon of hearing voices, it is possible to highlight how this experience is configured as a complex and multifaceted experience. It is also possible to point out how this is not a given reality but how it depends on the ways of knowing and the discourses put in place that help to define it [10,26,67,71]. In this sense, we can highlight the extent and impact on individuals’ biographies of traditional medical and pathologizing views that attribute to the experience the exclusive meaning of a prodromal stage or as a symptom of psychopathology, as well as on the medical professions involved. At the same time, much research highlights the scope and impact of other discourses with relevant pragmatic and economic spillovers on biographies and services. We can note, for example, how the meanings and use of the experience of hearing voices may differ in terms of risk or protective factors, rather than considered, for example, as more oriented toward high risk for psychosis [12,86]. This suggests the need to delve into the diversity of risk and protective factors for each person and situation, even when faced with hearing voices to grasp and assess needs most appropriately and effectively.

The basis of such reasoning, however, is not professional skills, but personal beliefs with the consequence of not being able to identify the possible resources present in their socio-family network.

Among the possible implications, therefore, it is possible to find the understanding of these experiences as primarily pathological, with a limitation of the vision of the development of potential resources through the existing varied family or service network and a drain on the resources of the only services considered relevant, namely medical and psychiatric services.

## 5. Conclusions

The present work has permitted the exploration of the modalities of the management of the phenomenon of hearing voices by family medicine physicians, the referral to another professional figure, and the evaluation of possible health interventions that can be activated. Indeed, their role is crucial because they are the first professionals whom the patient experiencing an unexplained psychological experience approach.

However, we have discovered a reductionist knowledge about the phenomenon, based on the idea that it is an undeniable sign of psychopathology, which causes serious consequences for the voice-hearers and those around them. Often doctors portray the voice-hearers as subdued by the voices, at the mercy of an extraneous and harmful presence that enters their life, which compromises their decision-making ability and can lead to aggressive and dangerous acts without being aware of them. These beliefs are represented through a process of generalization, based on negative stereotypes that are widespread in our society, regarding the subject at the expense of their experience improvements.

Thus, the research conducted here exposes the necessity of increasing the awareness of doctors and whole health workers about this phenomenon, in its variety of expressions and the possibilities of management. This highlighted the importance of embracing an attitude that is often open to new proceedings, that does not stop itself at what is already known to limit the risks by thinking and acting according to strict definitions of the past.

Releasing the health workers’ constitutions from pathological bias can also reduce the stigma experienced by voice-hearers, favoring the sharing of their experience and contact with normalizing and supporting sources.

Moreover, the recognition of the possible efficacy of other treatments, such as psychotherapy, could widen the range of interventions that can be activated, avoiding the need to resort electively to the administration of psychiatric drugs. All of this is to offer the person a service that guarantees a competent evaluation and an effective referral, calibrated on the case’s characteristics.

Broadening the view of possible experiences and meanings about the phenomenon of hearing voices, as well as the resources and impact of this in the biographies of hearers, can make it possible to consider from the earliest moments the orientation to interventions that can support the person in the most useful way. Indeed, this is made possible by tending to a health promotion aim, which sees each service, including primary care, as part of a network of services collaborating in this process. The risk becomes that of taking for granted the severity of certain experiences, with the implication of fueling the use of services to solve diseases rather than to promote health. In this light, instead, family physicians—as part of a health service network—can also take part and contribute in terms of resource development and health promotion.

## 6. Limitations and Strengths

A limitation of this research is that almost half of the participants are from a northeast region of Italy, and that may have given rise to context-influenced answers, even though the role of the family doctor is defined in national and not regional terms. Considering that this is one of the few research projects on this theme, the results of the research are exploratory, so it would be useful to augment the number of participants, maybe differentiating them by medical specializations.

Another limit concerns the fact that we have not deepened the territorial context in which the doctors are included, because, in some areas of the country, a greater number of services and resources could be useful to help people affected by mental illnesses.

Regarding the management of the specific interview, moreover, a few elements can be pointed out. This interview outline was formulated based on the specific objectives of the research, that is, considering general and exploratory questions with respect to the phenomenon and its management. The limitation was that of leaving in the background other issues that might be useful to investigate and specify.

## 7. Future Research

In connection with the research objectives and findings regarding the interview, further developments of the research could include the revision and reformulation of the interview outline, going to deepen and specify some peculiar issues. Among these could be included the investigation of the knowing references used by physicians to configure the phenomenon, the specific interventions fielded in these situations, and critical points and strengths detected by them.

Another aspect to be investigated with physicians could be that of any difficulties and coping strategies that they may observe on the part of their patients and in their family contexts. This investigation would allow us to gather elements not necessarily related to a psychopathological discourse, but to the specific biographies of the people involved. This would at the same time allow the research to use and develop these elements to contribute to expanding the physicians’ knowledge of the phenomenon and intervention from a perspective oriented to the complexity of the phenomenon and to health promotion.

It could also be investigated how relationships with other professionals, and specifically with psychiatric and psychotherapy services that might be involved in these situations, are told and managed. With this, additional theories, beliefs, and ways of handling such relationships could be brought to highlight specific elements that could be implemented and that could be a source of risk for physicians in their work in this field.

A further element of inquiry could concern questions or proposals made by the physicians with respect to what they might need in terms of knowledge, reference figures, and intervention tools to implement their knowledge and intervention methodologies, networking with other professionals.

Based on the above limits, it is considered interesting to expand the research conducted, involve more participants, and adopt systematic methods guided by this exploratory research. It would be useful to investigate how the phenomenon is configured and managed by other health professionals with whom auditors often come into contact.

In addition, it would be interesting to involve psychiatrists, nurses, and social workers in the research, because, from their evaluation, additional relevant data and insights could be gleaned. These professionals are indeed involved in the evaluation and management of the phenomenon under investigation. It could be useful to investigate how they configure the phenomenon and what criteria they use for assessment and intervention regarding specific role objectives.

Some of the doctors interviewed reported the lack of psychotherapists willing to take care of voice-hearers; it would be interesting to investigate whether, even in the case of psychologists and psychotherapists, the widespread prejudice exists that they are dangerous and excessively compromised people for whose support either a psychological or a psychotherapeutic intervention is indicated.

If so, this prejudice must be disavowed, starting with a greater focus on the power and implications of stigma and prejudice on the biographies and the professions involved. It might be useful to implement dedicated and supportive training courses and knowledge instruments for the so-called “experts of the psyche”. The inclusion of these contributions could allow for discussion and reflection with a view to the development of interventions that consider the role and context specificities, but that move in a network and according to shared modes of intervention.

## Figures and Tables

**Table 1 behavsci-14-00357-t001:** Demographic data.

	Gender	N	%
Male		18	51.4
Female		17	48.6
	**Age range**		
25–40		2	5.7
41–60		18	51.4
61–75		15	42.9
	**Experience Average as Family Medicine Physician**		
Male		22	62.8%
Female		21	60.0%

**Table 2 behavsci-14-00357-t002:** Track interview.

**OB.1—To investigate how the phenomenon of hearing voices is configured by the physicians, specifically what conceptions and beliefs are held about the hearers and their family members.**
How would you describe the phenomenon of hearing voices?
How do you consider hearing voices in relation to the lives of those who experience it and those around them?
**OB.2—To investigate how the management of a voice hearer occurs, focusing on intervention and referral to specialists**
As a clinician, have you ever had someone come to you and report hearing voices?
How did you decide to proceed?
May I ask what evaluation led you to make this decision?
If a similar situation occurred, how would you proceed?
Do you know of other types of intervention besides the one(s) you have chosen to activate?
In what cases might they be indicated? According to what evaluation and with respect to what needs?

**Table 3 behavsci-14-00357-t003:** How hearing voices and the voice-hearers are configured.

Processual Category	How Hearing Voices and the Voice-Hearers Are Configured	%
**Positionings**	Absolutizing the Hearing of Voices as a Psychopathological Symptom	85.7
	Diagnosing the Patient Based on a Hypothesis Derived from Personal Experience	68.5
	The Voice-Hearers as a Cause of Suffering for their Family	54.3

**Table 4 behavsci-14-00357-t004:** How the case intake and the referral to another specialist is managed.

Processual Category	How the Case Intake and the Referral to Another Specialist Is Managed	%
**Positionings**	The Exclusive Referral to the Psychiatrist	62.9
	The Autonomous Case Take-Over as a Symptom Containment Attempt	37.1

**Table 5 behavsci-14-00357-t005:** Referral Criteria.

Processual Category	How Referral Criteria Are Used	%
**Positionings**	Sending to the Specialist According to the Cause’s Origin: Physic or Psychiatric	45.7
	Referral to a Specialist based on the Idea of the Symptom Severity	37.1
	Sending the Patient to the Specialist according to their Age	17.1

## Data Availability

The dataset presented is not available for privacy reasons.

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
