# Peer review of "Early Support for People Who Hear Voices: Exploratory Research on Family Medicine Physicians’ Clinical Practice and Beliefs"

_behavsci, 2024, doi:10.3390/bs14050357_

Round 1

Reviewer 1 Report

Comments and Suggestions for Authors

This is a very interesting study focusing on the phenomenon of hearing voices, and its clinical implications. The paper is well written and of interest for the readers; however, several changes should be made before considering it for publication.

ABSTRACT

1-The introduction of the abstract should be more concise and focused on the hearing-voices phenomenon. Its has been described as a normal phenomenon but also as a prodromal stage and as a symptom of psychosis. These three scenarios should be stated.

2- The study design should be clearly mentioned in the abstract section. Is it qualitative? Phenomenological and exploratory study? Please, describe it with details.

INTRODUCTION

1- I recommend to propose different scenarios for this hearing-voices phenomenon. Clinical practice and interventions should be discussed in the context of these three scenarios.

2- Does DSM-5 really not report hallucinatory phenomena because it can be a normal phenomenon or because its consistency for the diagnosis of weak? Please, discuss more in depth this topic.

3- The main aim of this study should be described at the end of the introduction, not in the methods section.

METHOD

1- Theoretical Background should be described in the introduction section. Why are the authors conducting these study? That should be reported in detail at the end of the introduction.

2- Please, describe with more details the "snowballing technique" used to recruit the family medicine physicians.

3- Subsection 2.5. should be stated at the beginning of the methods section.

RESULTS

1- Regarding referrals to specialists based on symptom severy, I recommend, at the results section, to build a flow diagram according to the different scenarios and experiences of the family doctors.

DISCUSSION

1- What about discussing results based on findings on at risk mental states? This is highly needed in this paper.

"LIMITS"

This title should be changed to "Limitations and strenghts".

Author Response

We want to thank all reviewers for their comments and suggestions for modification and integration. We have used all suggestions to improve the clarity and argumentation of the article’s contents.

Reviewer 2 Report

Comments and Suggestions for Authors

Dears Authors

General comments

I think this is an article with significant content and some of interest.

In fact, the research conducted here exposes the need to increase the awareness of doctors and whole health workers about this phenomenon, which is a fundamental issue. Congratulations on the topic.

In my opinion, the article has various weaknesses which need to be reflected upon and possibly changed.

However, to make the article even better, I present my reflections according to your article.

 Title

Early Support for People who Hear Voices: Beliefs, Procedures, and Clinical Implications. Exploratory Research on Family Medicine Physicians” - Clear and directive

Abstract

- It's not clear what the aim of the study is. You should identify it.

Lines 23-26- You mention “We also discuss the importance of supporting clinical and psychotherapy pathways, oriented to the knowledge of the potential meanings taken on by the voices in the context of the personal and family history of the individual hearer.”- Not being part of the objective, it doesn't make sense to me.

- Keywords: why “stigma, clinical psychology” are not referenced either in the abstract or the title, why put it as a keyword? It doesn't seem appropriate. The direct objective of the work is not "health promotion", but an indirect objective. Could this be considered a keyword for your work? I have reservations.

Introduction

- Since 2022, the American Psychiatric Association Publishing, published DSM-5-TR (Psychiatry.org - DSM)- It makes no sense to publish an article in 2024 with the support of a manual with these characteristics from 2013, especially considering the object of study.

- I think your question is very little explored. In the DSM-5 that the authors refer to, it states "Diagnostic Issues Related to Culture - Socioeconomic and cultural factors should be considered, particularly when the individual and the clinician do not have the same cultural and socioeconomic background. Ideas that seem delusional in one culture (e.g. witchcraft) may be common in another. In some cultures, visual and auditory hallucinations with religious content (e.g., hearing the voice of God) are normal elements of religious experiences. (...) In some cultures, suffering may take the form of hallucinations or pseudo hallucinations and overvalued ideas that may present clinically similar to true psychosis but are normal to the patient's subgroup." Therefore, for your study to be clear, you should rely on the literature, explore more of what is referred to in the DSM-5-TR and with more diverse and reliable sources that explain the subject and report these benefits that the authors only identify briefly and little explored.

- Lines 78-82- “Some medicines do not have a good response rate, not even when the treatment is prescribed by a psychiatrist: for example, antipsychotics do not elicit a satisfying drug response in schizophrenic patients and cause at the same time disabling collateral effects, which in the long term seems to compromise irreversibly the social functioning in the person [35,36].”- You should support this information with other international sources, as well as Italian ones. In my opinion, it's essential to have good support to make this claim (which is, as you know, very debatable and still 'poorly' substantiated between costs/risks vs gains/benefits).

- Line 92- (N.764)- review typo

- Lines 97-107- You mention that you want to question: 1. “How do family doctors take charge of the voice-hearer patient?”; 2.”How do they interpret and conceive the hearing voices phenomenon?”; 3. “What do they give back to the person?”; and 4. “What type of health interventions do they know, and according to what criteria do they identify the most appropriate one?”. But you set yourselves goals: “This research aims to overcome the lack of information about this crucial point in the management of people who hear voices”. And then as another goal - “The aim has been to know the attitude of the primary care doctor toward voice-hearers, about the approach and the management of treatment to promote health. Indeed, we are interested not only in the choices for handling these patients but also in the conditions that establish these choices and the reflections that generate them.”. In my opinion, the text should be revised so that we agree on the questions and objectives. I believe that more than the content, the problem is related to the way it is written. I necessarily suggest a revision.

2. Method

2.1. This point does not exist. See typo.

2.2 Theoretical Background

- Lines 110-122- I don't think it's a supportive theory, but a comprehensive one. It is not, in my view, explanatory of the phenomenon under study. Re-evaluate.

2.3 Aims

- In the introduction you state your objectives: “The aim has been to know the attitude of the primary care doctor toward voice-hearers, about the approach and the management of treatment to promote health. Indeed, we are interested not only in the choices for handling these patients but also in the conditions that establish these choices and the reflections that generate them.”. In point 2.3 Aims, they mention 2 objectives: 1) To investigate how the phenomenon of hearing voices is configured by the physicians, specifically what beliefs and convictions are held about the hearers and their family members; 2) To investigate how does the management of a voice hearer occur, focusing on assessment and referral to specialists. There is no homogeneity. I don't understand that. In my opinion, it must be revised.

- Line 132- “Table 1. Track interview and aims”- does not appear referenced earlier in the body of the text.

- I ask, when you put as OB1, Lines 126-128 “…specifically what beliefs and convictions are held about the hearers and their family members.” How do you expect to evaluate it through the question, “How does hearing voices characterize the lives of those who experience it and those around them- if in the objective you put "beliefs and convictions" and in the question you ask about how to "characterize the lives of those who experience it and those around them?". I need clarification.

- I ask, when they put it as OB2, lines 129-130- “To investigate how does the management of a voice hearer occur, focusing on assessment a…”- Don't you think there's a question about knowing on what basis/foundations the doctor made the diagnosis?

2.4 Participants and Data Collection

- Line 148- “Table 2. Demographic data study participants.”- does not appear referenced earlier in the body of the text.

- Review how the data in Table 2. is presented in accordance with the Standards.

2.5 Methodology and Instruments

- Lines 150-153- “The research methodology used is qualitative [65]. In line with the goals and the theoretical background, the qualitative methodology makes it possible to collect discourses and analyze them through an interpretative process, to flesh out significant processes and the ways of constructing reality [66].”- I don't think it's necessary to post this information.

- Line 161-“…following the assumptions of qualitative research”- repetition.

- Line 166- where is the reference to 1 in the footer in the text? I can't find it.

-Lines 175- 176- “By meaning the discourse as practice-oriented to the situation (Potter & Edwards, 2001), we have chosen the…” - typo- review.

- Lines 186-188- “Once we found the fundamental discursive registers concerning positioning, we systematized the work-in procedural categories to share the meaning and facilitate the reading of the results”- it would be interesting for them to give a general overview of the procedural categories.

3. Results

3.1 How hearing voices and the voice-hearers are delineated

- It would be interesting, in my opinion, to present a paragraph with the actual results at the end of this point.

3.1.2 Diagnosing the Patient Based on a Hypothesis Derived from Personal Experience

- It would be interesting, in my opinion, to present a paragraph with the actual results at the end of this point.

3.1.3 The Voice-Hearers as a Cause of Suffering for his/her Family

- It would be interesting, in my opinion, to present a paragraph with the actual results at the end of this point.

3.2 How does the case take over and the referral to another specialist happens

3.2.1 The Exclusive Referral to the Psychiatrist

- It would be interesting, in my opinion, to present a paragraph with the actual results at the end of this point.

3.2.2 The Autonomous Case Take-Over as a Symptom Containment Attempt

- It would be interesting, in my opinion, to present a paragraph with the actual results at the end of this point.

3.3 Referral Criteria

- Line 301- The reference criteria don't make sense to me here, but they do in the place I pointed out above.

- I believe that the information contained in these sub-points should be included in the presentation of the results. In this way, the information appears disconnected.

- Line 313- where is the reference to 2 in the footer in the text? I can't find it.

4. Discussion

-Lines 386-389- You say “In other cases, indeed, given their evaluation, doctors consider the possibility of themselves temporarily managing the client by prescribing a pharmaceutical treatment. This data confirms some studies present in the literature [33, 77] that support the idea that an autonomous administration almost always has negative effects.”- I understand your reasoning, but what you are justifying with the literature is not your results. It is the possible results of this medical decision-making. It should be reviewed.

- Line 398- “cure process”- is that really what you mean?

- Line 414- You say “conducted in recent years [3, 37-48].”, but how recent? Reference 48 is from 1999 (25 years old); references 2 and 39 are from 2000 (24 years old); reference 42 is from 2001 (23 years old), 37 is from 2002 (22 years old).... and so on, how can you say recent?- Considero que a discussão poderia estar mais investida nos resultados.

5. Conclusions

- Line 448- “[24-28].”- references in the conclusion. It doesn't make sense to me.

- Line 472- “[11, 13].”- end the conclusion of your work with quotations? I don't think that's appropriate.

6. Limits

- In my view, not only the restricted location of the target population is a constraint, as is the small number, but above all the way in which the interview was formulated. I think you should reflect on this.

7. Future research

- Lines 496-498- You say “In that case, this stereotype needs to be disconfirmed, starting with greater attention to the power of stigma and prejudice”- stereotype or representation? Stigma and prejudice or symptoms? I think you should reflect on these questions. They start from an assumption that, in my opinion, may itself be biased.

- From my perspective, the lines you are proposing are far removed from your study objective. I invite you to think about challenges for future research that are more objective and aligned with your research.

References

- Review the references, there are several situations in which you are not in accordance with the journal rules, for example, where you put the date of publication.

Thank you for the opportunity to read and comment on your study.

I hope I have contributed to a reflection that can further improve your work.

I wish you good work.

Author Response

(The authors gave the same response as above.)

Reviewer 3 Report

Comments and Suggestions for Authors

I found the manuscript interesting and quite innovative.   I strongly agree with authors on the bias inherent in simplistically equating hallucination with psychosis. Unfortunatey this is the result of a certain type of psychiatry approach. Nowadays, most recent studies are highlighting that hearing voices is not necessarily a severe psychotic symptom. I'd invite Authors to implement the introduction with a brief overview of the most recent studies on the topic.  This would  help you to substantiate and enrich your work, making it more complete and understandable, especially for non-psychologist readers. You said "Numerous studies have shown that hearing- voices is not necessarily something pathologic, but it can also be functional for the person who experiences it [6,7]. Could you expand this statement? 

A second aspect I would highlight is presentation or results.  I think you shoud improve presentation and clarity of this section. I invite you to prepare a summary table of main results and to cut the text and aid the readability. 

Comments on the Quality of English Language

In my opinion, moderate editing of >English language should be done. 

Author Response

(The authors gave the same response as above.)

Round 2

Reviewer 2 Report

Comments and Suggestions for Authors

Dears Authors

General comments

I continue to consider that this is an article with significant content and of interest.

As I said before, the research conducted here exposes the need to increase the awareness of doctors and whole health workers about this phenomenon, which is a fundamental issue. I return to Congratulations on the topic.

In my opinion, the changes made were significant and made the article clearer, more objective, and more interesting. Congratulations.

However, to make the article even more robust, there are some improvements to be made and typos to be revised. I present my reflections according to your article.

Introduction

- Line 125- (N.264)- what does it mean? If it's the size of the sample, I suggest you put it according to the rules  (N=264)

- Not part of point 1.1. (Does not exist)- review

- Lines 128-131- “This difficulty and attempts by the doctors highlight the presence and the impact of these different conceptions about the phenomenon of hearing-voices, that is, the difficulty of managing the complexity of the phenomenon in terms of observation, evaluation, and intervention for what the person's needs may be in the face of the peculiar experience experienced.”- interesting phrase, but it is neither contextualized nor articulated with the information provided before/after.

2. Methodology

2.1. Participants

- Line 198- “Table 1. Demographic data”- does not appear referenced earlier in the body of the text.

- The percentages of the Experience Average as a Family Medicine Physician are missing.

2.2 Data Analysis

- Line 227- “Table 2. Track Interview”- does not appear referenced earlier in the body of the text.

3. Results

3.1 How hearing voices and the voice-hearers are delineated

- It doesn't seem clear to me to have a sub-point (3.1.) only illustrated with a table without being supported/framed.

3.1.1. Absolutizing the Hearing of Voices as a Psychopathological Symptom

- You should, in my opinion, present more concrete data regarding the positions. For example, absolutizing the Hearing of Voices as a Psychopathological Symptom (%); Diagnosing the Patient Based on a Hypothesis Derived from Personal Experience (%); The Voice-Hearers as a Cause of Suffering for their Family (%). For the reader to have a more complete perception/major framework.

Points 3.1.2.; 3.1.2; and 3.1.3

- In my opinion, you should provide more concrete data on the positions of the technicians.

3.2 How does the case take over and the referral to another specialist happens

- It doesn't seem clear to me to have a sub-point (3.2.) only illustrated with a table without being supported/framed.

Points 3.2.1.; and 3.2.2

- In my opinion, you should provide more concrete data on the positions of the technicians.

3.3 How Referral Criteria are used

- It doesn't seem clear to me to have a sub-point (3.3.) only illustrated with a table without being supported/framed.  

Points 3.3.1, 3.3.2, 3.3.3

- It doesn't seem clear to me to have a sub-point (3.3.) only illustrated with a table without being supported/framed.  

4. Discussion

- The point is missing (4.)

4/5. Conclusions

-Correcting a typo is not point 4, but point 5.

5/6. Limitations and strenghts

-Correcting a typo is not point 5, but point 6.

-No spacing at the beginning of the sentence. Revise.

- In my view, not only the restricted location of the target population is a constraint, as is the small number, but above all the way in which the interview was formulated. I still think that you should reflect on this and frame your reflection in the text.

6/7. Future research

-Correcting a typo is not point 6, but point 7.

- Lines 640-642- “In addition, it would be interesting to involve psychiatrists, nurses, and social workers in the research, because, from their evaluation, additional relevant data and insights could be gleaned.”- it would be interesting to give examples of data and insight.

- From my perspective, the lines you are proposing are still far removed from your study objective. I invite you to think about challenges for future research that are more objective and aligned with your research and to put them in the text.

 Thank you for the new opportunity to read and comment on your study.

I wish you good work.

Author Response

We want to thank the reviewer for the further comments and suggestions. We have used all the suggestions to improve the precision and argumentation of the article’s contents.
